# Xeric Tree Populations Exhibit Delayed Summer Depletion of Root Starch Relative to Mesic Counterparts

**Scott W. Oswald [1,2,*,†]** and **Doug P. Aubrey [1,2,†]**

1   Savannah River Ecology Laboratory, Aiken, SC 29802, USA; daubrey@srel.uga.edu
2   Warnell School of Forestry and Natural Resources, University of Georgia, Athens, GA 30602, USA
*   Correspondence: swo57118@uga.edu; Tel.: +1-803-725-0070
†   These authors contributed equally to this work.

**Abstract:** Research linking soil moisture availability to nonstructural carbohydrate (NSC) storage suggests greater NSC reserves promote survival under acute water stress, but little is known about how NSC allocation responds to long-term differences in water availabilty. We hypothesized populations experiencing chronic or frequent water stress shift carbon allocation to build greater NSC reserves for increased survival probability during drought relative to populations rarely experiencing water stress. Over a year, we measured soluble sugar and starch concentrations from branches, stems, and coarse roots of mature *Pinus palustris* trees at two sites differing in long-term soil moisture availability. Xeric and mesic populations exhibited a cycle of summer depletion-winter accumulation in root starch. Xeric populations reached a maximum root starch concentration approximately 1–2 months later than mesic populations, indicating delayed summer depletion. Xeric and mesic populations reached the same minimum root starch at similar times, suggesting extended winter accumulation for xeric populations. These results suggest seasonal mobilization from root starch is compressed into a shorter interval for xeric populations instead of consistently greater reserves as hypothesized. Seasonal trends differed little between xeric and mesic populations for starch and sugars, suggesting the importance of roots in seasonal carbon dynamics and the primacy of starch for storage. If roots are the primary organ for longterm storage, then our results suggest that whole-plant mobilization and allocation respond to chronic differences in water availability.

**Keywords:** nonstructural carbohydrates; stored carbon; carbon allocation; drought; *Pinus palustris* (longleaf pine)

---

## 1. Introduction

The survival of many tree species depends on stored carbohydrates or nonstructural carbohydrates (NSCs; e.g., sugars, starch) [1,2]. Seasonal asynchrony of photosynthesis (energy supply) and growth, reproduction, or respiration (energy demands) suggests that many survival strategies involve storing carbohydrates (and therefore storing free energy) [3]. A number of studies observe evidence suggestive of carbon allocation to storage preferentially over growth [3,4]. Several studies have observed changes in NSCs in response to disturbances such as fire [5–7] and frost [8]. While the underlying controls on storage remain poorly described [9], storage is a critical component of plant stress response.

Greater allocation of NSCs and associated energy to storage could provide resilience to water stress [10–12]. Plants respond to water stress by ceasing growth [13–15] and reducing stomatal conductance to prevent hydraulic conductance loss via cavitation [16,17]. Both responses impact NSC production via photosynthesis and NSC consumption by growth [18,19]; however, growth

cessation occurs at higher soil water potentials before significant reduction in photosynthesis [14,20,21] with documented increases in NSCs under mild water stress [22,23]. NSC reserves provide an energy supply buffering metabolism under reduced photosynthesis [24]. Studies also link NSC allocation directly to hydraulic function via the embolism repair hypothesis in which concentration gradients of soluble sugars permit the refilling of embolisms under tension [25,26], alleviating the loss of hydraulic conductance for xylem [16,17]. Both photosynthetic/growth losses and hydraulic conductance losses are factors in drought mortality [27], possibly depending on drought strategy (i.e., anisohydry or isohydry) [28]. Several studies document decreases of NSCs, especially root starch, under drought [11,29] while others document maintenance of NSCs [30,31] or a combination of the previous responses [32], suggesting NSC allocation has a complex relationship with drought survival, and the exact physiology of drought mortality is still unresolved [10,33,34]. Regardless, storing NSCs before drought may confer greater survival during drought, and some studies directly link drought survival or recovery and NSC reserves [11]. Therefore, long-term low water availability may encourage storage in anticipation of water stress.

Many studies observe different NSC dynamics across organs [35–38]. These differences are an important factor in explaining the different observed NSC responses to drought. Several studies present evidence suggesting NSC responses differ by organ [28,35,39]. Although more difficult to measure, root starch exhibits a high degree of temporal variation [24,36], and root starch variation is linked to drought survival [28,29]. Intense water stress can cause shoot death by inducing root embolisms [40], pointing to the importance of root NSC reserves for repairing embolism to prevent dieback. Shoot reserves could serve primarily to maintain xylem (through embolism refilling [41] or via osmotic adjustment [42]) and phloem transport [43]. Different roles of sugar and starch across organs suggests that environmental stimuli may perturb NSC dynamics differently in different organs, therefore it is critical to consider both the root and shoot systems. Contrary to expectations, stem variation is low in previous observations [36,44,45].

The links between drought and NSC dynamics suggests that greater NSC reserves promote survival under acute water stress. Thus, we hypothesized that greater NSC reserves would be observed in populations that often experience water stress. Few studies have considered the impact of long-term water availability on seasonal NSC dynamics (but see [31]), except in studies of delayed drought mortality [11,46]. We present here observations of NSC differences between mature, wild *Pinus palustris* (longleaf pine; Mill.) experiencing long-term, chronic differences in soil moisture availability due to soil drainage differences. We selected two geographically close stands in a *P. palustris* woodland, a xeric site with well-drained soil and a mesic site with poorly drained soil. The sites were chosen to minimize all other abiotic and biotic environmental differences between the populations, creating a natural experiment in which the xeric population experiences greater soil drainage than the mesic population. Their proximity also suggests these populations are not reproductively isolated and therefore not genetically separate. Differences in these particular populations have been studied for >20 years and the soil drainage differences generate consistent differences in productivity [47–58]. We measured sugar and starch concentrations from lateral root, main stem, mid canopy branch, and upper canopy branch tissues for a year, and we compared the temporal variation of these concentrations to infer differences in carbon allocation. If the capacity to deplete NSC reserves during drought confers greater survival, then we expected to observe higher root starch concentrations over time for the xeric population than the mesic one as a result of greater storage during non-drought years.

## 2. Materials and Methods

### 2.1. Study Site Description

Our study was conducted in a longleaf pine (*Pinus palustris* Mill.) woodland ecosystem located at the Joseph W. Jones Ecological Research Center at Ichauway, Baker County, Georgia, USA (31°13′14.3″ N, 84°28′42.9″ W, 48 m above sea level). These *P. palustris* woodlands have been

managed for more than eight decades with frequent prescribed fire [51]. The region's climate is humid subtropical [24]. Based on the period 1981–2010, annual precipitation at the site is 1400 mm, mean annual temperature is 18.6 °C, with mean annual maximum temperature of 25.2 °C and mean annual minimum of 11.9 °C [51].

We selected a mesic site and a xeric site along an edaphic gradient based on soil properties. The mesic site is poorly drained sandy loam over sandy clay loam or clay textured soils, classified as Aquic Arenic and Typic Paleudults. A clay-textured lens decreases water infiltration into deeper soil (argillic horizon within 0–95 cm of surface). Soil mottling within 0–30 cm indicates poor drainage, and significant rainfall events often leave standing water. Located on upland sand ridges, the xeric site was well drained with sandy soils exceeding 2.5 m in depth, no argillic horizon, weak horizon development, and low organic matter content. Its soils are primarily classified as Typic Quartzipsamments with inclusions of Arenic or Grossarenic Kandiudults [51]. Other than soil hydrology, both sites experience similar light, temperature, and vapor pressure environments (Figure 1 in [54]; Figure 2 in [57]). Xeric site annual precipitation was 94% of mesic site annual precipitation for 2008-2015 (Table 1 in [55]). Soil volumetric water content was 1%–3% higher at the mesic site during the study period and the preceding five years (Figure 1 in [53]; Figure 2 in [57]). The hydrological differences resulted in stark differences in net ecosystem exchange ($-208.2$ and $-73.7$ g $CO_2$ m$^{-2}$ year$^{-1}$ for the mesic and xeric site respectively; Table 1, Figure 1, and Figure 2 in [55]; Figure 3 in [57]). Furthermore, the xeric population of *P. palustris* has lower sapwood area index (SAI), root area index (RAI), leaf area index (LAI), sapwood-leaf area ratio (Huber ratio), and root-leaf area ratio; but similar overall conductance to the mesic population [50]. For these reason, numerous other studies have employed this edaphic gradient to study ecosystem carbon dynamics [55–57].

## 2.2. Tissue Sampling and Chemical Analysis

We repeatedly sampled the sugar and starch concentrations in the coarse roots (diameter > 2 mm) and stem of five mature, canopy *P. palustris* individuals at each site. To avoid too much disturbance of dynamics from destructive sampling, we measured the sugar and starch concentrations of the upper and mid canopy branches from a second set of five *P. palustris* individuals. We sampled stems by coring the main trunk at a height of 1.4 m. We sampled roots by tracing and cutting a root from the selected tree. We sampled stems monthly from April 2013 to March 2014 while we sampled roots at two-month intervals from March 2013 to January 2014. We sampled branches at two-month intervals from June 2013 to April 2014. Populations are likely not reproductively isolated given their geographic proximity, that *P. palustris* is wind pollinated, and that both sites are contained within a larger matrix of longleaf woodland vegetation with many *P. palustris* individuals.

Immediately after collection, we cleaned tissue samples with deionized water and cooled them to 0 °C for at least 48 h to stop metabolic activity. We then dried the tissue samples at 65 °C for 72 h. After drying, we ground tissue samples with a ball mill (Spex Sampleprep 8000D, Spex Centiprep, Metuchen, NJ), and collected ground tissue in a 20 mL plastic scintillation vial. Prior to extraction and quantification, we dried ground samples again at 70 °C for another 48 h to ensure all moisture was removed. Sugars were extracted using ethanol and water mixture; after sugar extraction, starch was digested using sulfuric acid. We used a modified sulfuric acid-phenol method to quantify soluble sugars and digested starch given in [59,60]. We report all concentrations as a mass fraction per mil (mg g$^{-1}$).

## 2.3. Data Analysis

We analyzed sugar and starch concentrations separately due to different physiological interpretations. Furthermore, low correlation between sugar and starch concentrations ($r = -0.04$) indicates that estimates for each carbohydrate provide little statistical information for improving the estimate of the other carbohydrate. Sugar and starch concentrations from each organ are analyzed separately, because each organ was measured at different time intervals, with different starting and

end dates, and because branch tissue samples also came from a different set of trees than the stem and root tissues. To compare sugar and starch variation within each organ across site while controlling for temporal variation, we constructed a generalized-additive-model (GAM) assuming uncorrelated, Gaussian error. Briefly, the temporal variation within each site-organ is estimated as a smooth function (Equation (1)) where *j* ranges over sites and organs and *i* over the data.

$$y_i = \alpha_j + f_j(x_i) + \varepsilon_i \tag{1}$$

The smooth functions were estimated using ten penalized thin-plate regression splines [61] with restricted maximum likelihood for fitting [62] (an identifiability constraint removes one degree of freedom giving a rank equal to nine). For parameter estimation, we used restricted maximum likelihood (REML) employing the software implementation provided by mgcv package (version 1.8–31) [62,63] in R (version 3.6.3 "Holding the Windsock") [64]. We used approximate Wald statistic tests for testing the absence of temporal dynamics (with a rank-reduction based on the effective degrees of freedom) [65] and a Wald statistic test for testing differences in estimated smooths between sites based on the Bayesian posterior covariance for the estimated parameters (without the rank-reduction) [62,66]. Confidence intervals and bands were derived from the Bayesian posterior covariance [62,66]. To compare maxima and minima of estimated smooths, we estimated the optima using Brent's method (golden section search with parabolic interpolation) [67] using the the software implementation provided by R [64]. To quantify uncertainty in estimated optima, we used $\mathrm{Var}[x^*] \sim \frac{\partial x}{\partial \beta}(x^*) V_\beta \frac{\partial x}{\partial \beta}^T(x^*)$ denoting the optima as $x^*$, where the derivative $\frac{\partial x}{\partial \beta} = -\left(\frac{\partial^2 \mu}{\partial x^2}\right)^{-1} \frac{\partial \mu}{\partial x \partial \beta}$ is estimated as an implicit function based on the first order optimal condition $\partial \mu / \partial \beta = 0$. We only consider optima within the study period and not at or beyond the boundary. For each null hypothesis significance test, we report the statistic and p-value (degrees of freedom indicated by subscript) except for significance tests for the estimated smooths where the highest *p*-value is reported only. All confidence intervals are at the 95% level.

## 3. Results

### 3.1. Starch

We observed significant temporal dynamics for starch concentrations (all $p \leq 0.011$) in all organs, but we only observed a seasonal cycle of winter accumulation and summer depletion—with a clear maximum and minimum—in roots (Figure 1a). We observed distinct root starch temporal dynamics between xeric and mesic populations ($\chi^2_9 = 24.4$, $p = 0.0038$). To summarize the differences in temporal dynamics, the accumulation of the xeric population lasted one or two months longer than mesic population (Figure 1a). We did not detect a significant difference in maximum or minimum concentrations (maximum $179 \pm 25.6$ mg g$^{-1}$ and $151 \pm 22.7$ mg g$^{-1}$ for xeric and mesic, respectively; $Z = 1.598, p = 0.11$; minimum $31 \pm 31.7$ mg g$^{-1}$ and $19.5 \pm 25.2$ mg g$^{-1}$ for xeric and mesic, respectively; $Z = 0.5566, p = 0.58$). We did detect a significant difference in the time-of-maximum (argmax) root starch between xeric and mesic populations; the argmax for xeric *P. palustris* occurred $1.6 \pm 0.01$ months after the maximum for mesic *P. palustris* (June-July 2013 and April-May 2013 respectively; Figure 1a; $Z = 146$). We estimated xeric and mesic minimum (argmin) occurred approximately at the same time (October 2013 for both; Figure 1a; difference $0.2 \pm 0.03$ months), indicating the summer depletion period was shorter for xeric *P. palustris* than mesic *P. palustris* (roughly 3.5 months vs. 5.4 months respectively). In other words, we observed differences in timing of when xeric and mesic populations switch from accumulation to depletion of root starch (Figure 1a). We did not detect differences in the temporal dynamics in branches or stems (Figure 1b–d).

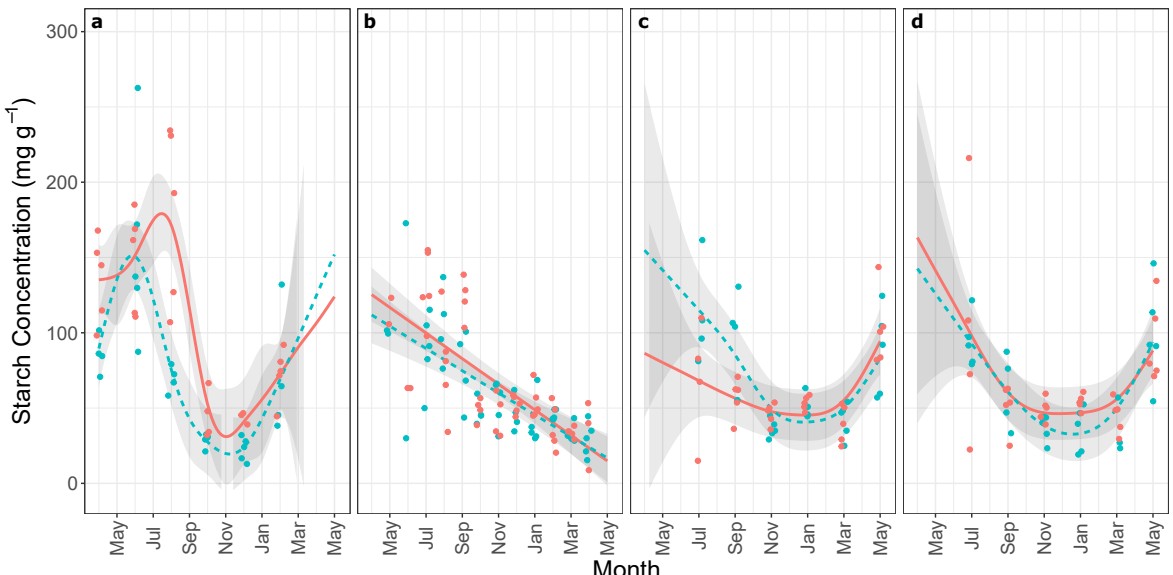

**Figure 1.** Mean starch concentrations (and 95% confidence band) from xeric (red solid) and mesic (blue dashed) populations of *Pinus palustris* collected between April 2013 and May 2014 from coarse lateral roots (**a**), stem at breast height (**b**), mid canopy branches (**c**), and upper canopy branches (**d**); estimated by generalized additive model (ten thin-plate regression splines using REML). For visibility, data points are jittered ±0.2 months along the x-axis (jittering not used in fitting).

### 3.2. Sugars

Unlike starch, we observed differences in the average sugar concentration over time ($F = 116, p < 10^{-16}$), in order from lowest to highest: stem, mid canopy branches, upper canopy branches, roots. We did not detect a difference between upper and mid canopy branches ($t_{233.9} = 1.79, p = 0.075$). We observed significant temporal dynamics for sugar concentrations (Figure 2; all $p < 0.048$), except in stems (Figure 2b; $p = 0.81, 0.41$ for xeric and mesic respectively). We observed distinct temporal dynamics in root sugars ($\chi^2_9 = 32.7, p = 0.00015$) but not in any other organ (all $p > 0.26$). Except for stem sugars, we observed higher winter sugar concentrations (Figure 2). We did not observe a clear seasonal cycle of accumulation-depletion in root sugars. Even with smoothing, we observed two maxima for mesic root sugar concentrations and only one for xeric roots (Figure 2). The lack of a seasonal cycle along with the considerable spread in observations suggests that further analysis would be particularly suspect and unreliable as these differences might not reflect differences in seasonal dynamics. Therefore, we did not analyze this temporal difference further.

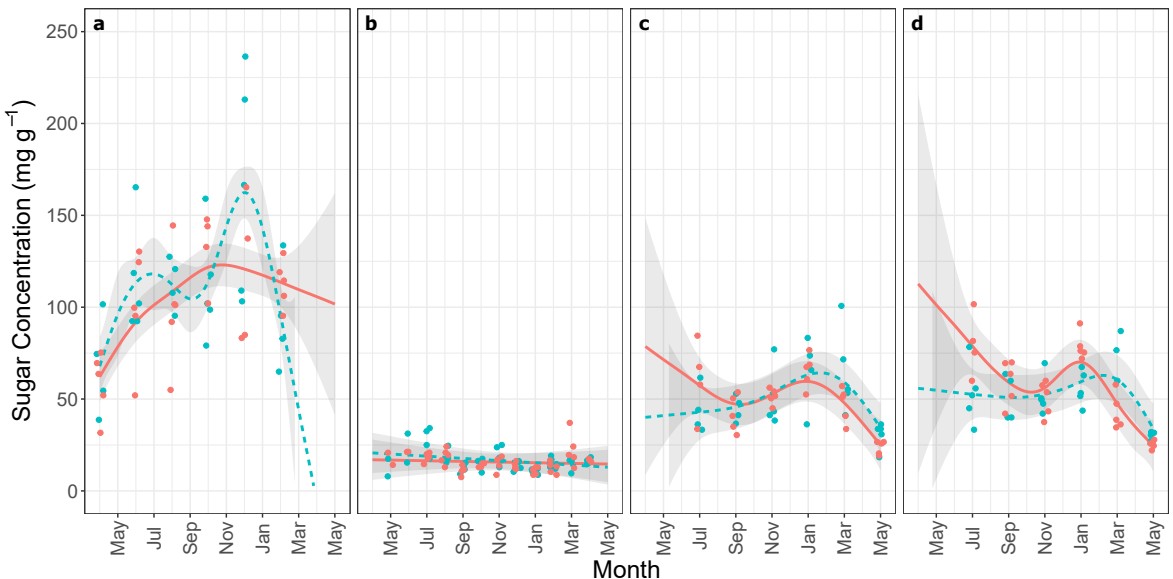

**Figure 2.** Mean soluble sugar concentrations (and 95% confidence band) from xeric (red solid) and mesic (blue dashed) populations of *Pinus palustris* collected between April 2013 and May 2014 from coarse lateral roots (**a**), stem at breast height (**b**), mid canopy branches (**c**), and upper canopy branches (**d**); estimated by generalized additive model (ten thin-plate regression splines using REML). For visibility, data points are jittered ±0.2 months along the x-axis (jittering not used in fitting).

## 4. Discussion

Both xeric populations accumulated root starch for one to two extra months, compared to mesic populations (Figure 1). This observation provides evidence that carbon allocation responds to long-term differences in soil water availability. It reinforces the link between NSC physiology and drought survival in woody plants [3] and suggests plastic allocation is possible. We highlight three aspects of our finding. (1) We did not expect the distinct temporal dynamics but consistently higher concentrations over time for xeric populations. Instead, our findings suggest that accumulation-depletion dynamics differ in seasonal distribution but not overall magnitude for xeric and mesic populations. (2) Observed dynamics differ for sugar and starch concentrations. Starch concentrations exhibit greater temporal variation, consistent with seasonal oscillation; we observed the clearest differences in starch concentrations. (3) Observed dynamics differ across organs. We observed temporal variation for stem and root starch concentrations, but only for root starch did temporal dynamics differ between xeric and mesic populations, suggesting root starch is the primary form and location for longterm storage in these trees similar to herbaceous plants [36,68,69]. Our results have several limitations. First, inferences about temporal dynamics are sensitive to temporal resolution. We measured root NSC measurements at two-month intervals, limiting our ability to detect fine-scale changes. Second, we only observed a single annual period, limiting inferences about interannual seasonal dynamics. Comparison with other data from the same species at similar sites [24,44] and of NSC seasonal variation in general [3,36] suggests that seasonal variation observed here is consistent through time. We collected the data in a particularly wet year (See site rainfall data in [57]), suggesting these differences do not result from acute drought responses. However, both sites experienced drought in the two-three years prior to measurement (−5 to −2.5 Palmer Drought Severity Index; See [57]). NSC measurements also have methodological limitations [70]. We repeated our chemical quantification method if replicates from the same tissue sample differed by more than 10% to reduce the noise inherent in these measurements. As noted earlier, root and stem measurements originate from a different sample of trees than branch samples limiting comparisons across organs.

Overall, sugar concentrations exhibited few differences in dynamics between xeric and mesic populations. We observed small differences in temporal dynamics within branch sugar concentrations in which xeric populations switched from winter accumulation to spring-summer depletion earlier than mesic populations (Figure 2). Root sugar concentrations increased during winter possibly as cold-weather adaptation (perhaps by freezing point depression) [8]. Freezing point depression is a possible explanation for branch sugar dynamics but does not explain differences in xeric and mesic dynamics. This difference could also result from the same processes which generated the differences in root starch, although a meta-analysis of the seasonal dynamics of leaf and stem sugar concentrations across many conifers did not reveal a pattern of winter-accumulation and summer depletion similar to the cycle observed in belowground organs [36]. Our observations of stem and branch sugars is similar with observations in the deciduous *Pistacia vera* (pistachio) and *Juglans regia* (walnut), in which stems exhibited little temporal variation but branches and twigs did; but not *Prunus dulcis* (almond) in which stems and branches exhibited similar temporal variation (roots were not measured) [38]. Previous observations of the same site also found seasonal variation in root sugar and starch, but not in stems [44] similar to observations in other systems [45]. It is also possible that xeric and mesic populations had different dynamics in specific sugar compounds (e.g., sucrose, glucose, fructose, etc.) as observed in other studies [69,71,72] that we were unable to detect.

Our results suggest plasticity in NSC allocation. However, we did not measure plasticity directly and do not know if observed differences result from different environments over the life history of the two populations (i.e., ontogenetic/development of seedlings in different conditions) or from recent events (i.e., carbon dynamics responding to changing environmental conditions without changing allocation priority) [73]. The differences in dynamics should not result solely from genetic differences because the populations are not reproductively isolated given their geographic proximity. Both sites are contained within a larger matrix of longleaf woodland vegetation with many *P. palustris* individuals, and *P. palustris* is wind pollinated. However, it is possible that genotype filtering occurs early in the life cycle and therefore creates distinct genetic populations. Genetic differences have been associated with differences in NSC concentration [74].

The implications of differences in temporal dynamics depend on which processes are responsible. Differences in assimilation, growth, osmoregulation, and respiration drive these dynamical differences, but the relative contribution of each process changes the interpretation of results. The sink limitation hypothesis suggests downregulation of growth due to differences in water and temperature during the spring and summer [21,75]. However, differences < 1°C in monthly average temperature between the sites (See [57] for detailed temperature data) renders temperature sink limitation unlikely. Soil volumetric water content differed by 2%–3% for this period but does not mirror the temporal dynamics here (See data in [57]). Therefore, water sink limitation is possible but seems unlikely to generate observed results. More precise data on soil and plant water potentials or hydraulic conductance is not available for the study period. NSC accumulation for osmoregulation is another possible explanation [76,77]. None of the observed NSC dynamics mirror soil volumetric water content for that time period (See data in [57]), suggesting that observed dynamics do not solely reflect changes in osmotic potential. Another possibility is that the observed differences reflect a drought recovery period given that carbon allocation and NSC depletion are linked to delayed drought mortality and recovery [46]. Both xeric and mesic populations experienced drought between 2011 and 2012 before the study began (See data in [57]). These differences in temporal dynamics without mirrored differences in current hydrologic conditions suggest short term hydrologic differences did not result in the observed patterns. We suspect the presence of active regulation of NSC allocation in response to the long-term hydrologic conditions [3]; however, further observations are needed to confirm this conjecture. Our measurements suggest that seasonal differences may be subtle and difficult to detect in longterm studies with low frequency measurements [31].

We speculate that delay in root starch depletion reflects a conservative "wait-and-see" strategy. Xeric trees delay growth in case drought conditions occur, growing only if early summer is particularly

wet. When drought occurs, extra reserves maintain metabolism, hydraulic function, and cell hydration [24,26]. Where droughts are rarer, competition for other resources like light and nitrogen dominates provide the advantage to less conservative allocation strategies. Allocating to growth early maximizes growing season photosynthetic capacity by increasing the total leaf area. If the observed allocation results from a balancing act between light and nutrient competition which requires fast growth, favoring quick NSC investment, and drought/disturbance resilience which requires building starch reserves, favoring storage [69]. A similar balancing act is seen with carbon-nitrogen allocation strategies [78,79].

## 5. Conclusions

We observed a delayed summer increase in the root starch concentrations for xeric population of *P. palustris* compared to a similar mesic population, suggesting differences in seasonal storage due to long-term differences in hydrologic conditions. We did not observe clear differences in other organs or in sugar concentrations, suggesting the primacy of root starch in seasonal carbon storage. Our results also highlight the complexity of NSC dynamics and the necessity for more precise predictions from theory and models of theory. Greater precision in prediction permits more informative experiments and allows for more fully experimental confirmation of models. Achieving these goals strengthens predictions of ecosystem response under climate change, where predictions cannot be experimentally verified. Future research should aim to evaluate seasonal NSC dynamics in additional species in sites with contrasting soil moisture availability over multiple years to confirm or challenge our finding as well as to explore NSC reserves respond to chronic and internanual variation in soil moisture availability.

**Author Contributions:** Both authors contributed equally. Both authors have read and agreed to the published version of the manuscript.

**Funding:** Funding was provided by the Robert W. Woodruff Foundation through the Jones Center at Ichauway. This work was also supported by the Department of Energy under Award Number DE-EM0004391 to the University of Georgia Research Foundation as well as USDA National Institute of Food and Agriculture, Agriculture and Food Research Initiative McIntire Stennis project 1023985.

**Acknowledgments:** The authors thank A. Baron and M. Drews for their assistance with data collection, logistics, and laboratory analyses.

**Conflicts of Interest:** The authors declare that they have no conflict of interest.

## Abbreviations

The following abbreviations are used in this manuscript:

NSC    nonstructural carbohydrates
GAM    generalized additive model
REML   restricted maximum likelihood

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
