# Peer review of "Xeric Tree Populations Exhibit Delayed Summer Depletion of Root Starch Relative to Mesic Counterparts"

_forests, doi:10.3390/f11101026_

Round 1

Reviewer 1 Report

The work of Oswald and Aubrey is an interesting analysis of differences in the seasonal evolution carbohydrates pool in two P. palustris populations. The populations represent two contrasting degrees of historical exposure to drought periods. The work is timely and very interesting. However, my main concern is with the sampling protocol. Authors make use of different target trees to sample stem and roots regarding upper and mid-canopy branches. This seems to me not very correct and could undermine the soundness of their conclusions. In my opinion, it’s a pitfall to be considered before a final decision on the acceptance of the paper. Maybe, an explicit recognition by authors of the limitations of their work in the material and methods section, before the comments included in the discussion, could be enough to warn potential readers about this issue. On the other hand, the work is interesting.

Abstract

Most of the summary is focussed on the seasonal changes of starch. Please, let’s include some comments about soluble sugars. Besides, you pay more attention to the patterns in roots. What about the trends in other organs? To provide support to your conclusion about “mobilization and allocation respond to chronic differences in water availability” you have to provide some information relating the seasonal changes in roots with those from other organs.

Introduction

The introduction is written clearly. Maybe, I miss some more references that would consider specifically the behavior of conifers. The bibliography is abundant in this matter.

Page 1 line 19 -27 I suggest authors focus on species from conifers. They seem to show different patterns to Angiosperms in the dynamics of the carbohydrates pool. The literature is abundant in examples in the last years. I provide you with some references on the matter, some of them very recent:

Klein T, Hoch G, Yakir D, Korner C (2014) Drought stress, growth, and nonstructural carbohydrate dynamics of pine trees in a semi-arid forest. Tree Physiol 34:981–992.

Urrutia-Jalabert R, Lara A, Barichivich J, Vergara N, Rodriguez CG and Piper FI (2020) Low growth sensitivity and fast replenishment of non-structural carbohydrates in a long-lived endangered conifer after drought. Front. Plant Sci. 11:905.doi: 10.3389/fpls.2020.00905

Liu H, H Shangguan, M Zhou, P Airebule, P Zhao, W He, Ch Xiang, X Wu (2019) Differentiated responses of nonstructural carbohydrate allocation to climatic dryness and drought events in the Inner Asian arid timberline. Agricultural and Forest Meteorology 271:355-361.

Brunner I, Herzog C, Dawes MA, Arend M, Sperisen C (2015) How tree roots respond to drought. Front Plant Sci 6:1–16.

Page 2 line 51 This function can be considered for species with re-sprouting capacity. This is not very often the case for most conifers. Try to qualify the statement by rewording the sentence.

Page 2 line 55 Include some more references on the low carbohydrates variability in stems of pines as observed recently in the following paper:

Gea-Izquierdo, G, Aranda, I, Cañellas, I, Dorado-Liñán, I, Olano, JM, Martin-Benito, D. Contrasting species decline but high sensitivity to increasing water stress on a mixed pine–oak ecotone. J Ecol. 2020; 00: 1– 16. https://doi.org/10.1111/1365-2745.13450

Page 2 line 63 Try to note populations were so nearby, that genetic differences were not expected, and most of the observed changes were a consequence of the phenotypic plasticity in response to water stress endured by both populations along the sampling years.

Page 2 line 70 The use of “deplete” is actually in line with your argument? Following your previous comments in the introduction, it seems more logical to raise the hypothesis that the capacity to building up of carbohydrates pool in populations submitted historically to drought is what would be expected to confer higher resilience to those dry prone populations –as a working hypothesis-.

Material and methods

Line 103 I think this point is one of the most remarkable shortcomings of the work. Why were different the trees for sampling aboveground and belowground organs? You should have sampled on the same trees to have analyzed in deep and correctly the seasonal interplay between starch and soluble sugars.

Line 116 Why did you not express it with other units? gr gr-1 dry weight or mol gr-1 dry weight. They are most often used in the literature about carbohydrates in plant tissues. This would help to make it easier to compare your results with those from others.

Results

Line 147 Rephrase the sentence. It’s not clear: “between for” ???.

Minor comment: The quality of graphs could be improved. Include the name of months instead of numbers in the “x” axis, with a two months separation at most.

Line 164 – 166 I do not see this difference. There are no seasonal changes in the carbohydrates of stems, neither for pines in wet or dry sites. I think that you can only conclude something about differences among organs, but without any clear seasonal trend and differentiation between the two sites.

Discussion

The sampling in different trees, as already commented, limits the possibility to discuss the interplay of changes in starch vs. soluble carbohydrates. In my opinion, some of the unobserved differences in carbon allocation, common on other similar works (e.g. seasonal remobilization of carbohydrates among organs, or opposite evolution of starch vs. soluble carbohydrates at some times of the growing season) could rise from this shortcoming in the sampling. It’s difficult to establish a credible story about the seasonal change of both pools of non-structural carbohydrates. Probably the most puzzling result was the absence of a clear seasonal change in the soluble carbohydrates pool. In summary, I see it difficult to connect both kinds of results into a compressive and consistent discussion.

Lines 197-207 This paragraph should be moved to the beginning of the discussion. You must warn potential readers about the shortcomings of your study. You should add as a limitation the sampling of roots and stems in some trees, and upper and middle branches from other different ones.

Line 212-213 Include a comment about this point in the material and methods section.

Line 224 Source instead of sink???

Line 227 – 228 This is a general statement without a clear relationship with your study. Besides, you include for instance issues that are controversial and not observed as general patters (e.g. refilling of vases from carbohydrates). The discussion is littered with similar sentences referring to well-known general physiological processes but without a clear link with your work.

Line 236 This conclusion would have required a longer temporal series to be supported. Tune down your comment, and recognize this limitation of the study. You are working with almost one unique year to raise any conclusion about patterns in the long-term.

Line 242 Allocation at most would maximize the seasonal carbon uptake of the tree, but it is not clear to me how can maximizes photosynthetic capacity –this terminology is reserved normally to individual leaves-. I have observed that some other scientific concepts are not used with precision throughout the main text, and in particular in the discussion.

Line 242 – The following sentence: “The observed allocation is a balancing act between light and nutrient competition (favoring quick NSC investment) and drought/disturbance resilience (favoring NSC storage)” is an interesting appreciation by the authors. However, I had to read a couple of times to understand. Try to rephrase to improve a bit the understanding.

Line 248 “Sink” better than “pool”???When referring to allocation processes it is common to consider organs in terms of source or sinks for resources as carbon.

Minor comment: Line 250 Include the name of the first authors at the beginning of the sentence. That is: Schönbeck et al. [30] sampled NSC dynamics…

Line 245 -255 The overall paragraph relies on the discussion of the work by Schönbeck et al. I think the main subject has to be discussed at the view of more papers. Review the bibliography, and try to elaborate a discussion confronting your results with more papers than that from the aforementioned authors.

Line 255 – 268 The full paragraph contributes little to the general discussion. It’s a bit verbose as mention very general issues marginally related to the main points of the paper. It could be eliminated, or at least shortened, without any serious impact on the general discussion.

Minor comments

The affiliation 3 is missing for one of the authors. The symbol depicting both authors contributed equally to the manuscript I think should be included after the name of the authors as the rest of the symbols.

Reference 21 The name’s journal and year are missing from the reference. Similar lacks and errors are observed in other references (e.g. year publication in 39, 40). The overall bibliography should be carefully reviewed by the authors.

The graphs are of very low quality and represent almost raw graphs from the use of the ggplot2 package of R. I’m sure authors can do an effort to improve the aspect of the figure for publishing in a scientific journal.

Author Response

Please see the attachment to view the point-by-point responses to the comments.

Reviewer 2 Report

The subject of the work is interesting because it is very little information available on the effect of chronic or frequent water stress (drought stress) on accumulation and allocation of  carbohydrates in trees.

Major comments concern Materials and Methods

and the presentation of the results and their description.

Line 9:  approximately 2  months (it is shown in the Figure 1)

Line 10:  the research concerned only one species (not in both species)

Line 75-76: The study was conducted in a longleaf pine (Pinus palustris Mill.) was not there  wiregrass (Aristida stricte  Michx.)

Line 78-99: The description of the research site is correct, however, frequent references to the publications are compiled. it would suffice to quote the authors of one publication in line 89 and to resign from the other earlier publications. This also applies to line 90-99 and referring to numerous figures and tables. Information in this field should be transferred to the introduction.

Line 105: From the selected tree (on what basis were the trees selected, the physiological age, or other criteria, or was it chosen at random?)

Line 140: starch Line: 142: the accumulation of the xeric population lasted approximately two months.Figures 1 and 2 should be divided into A, B, C, D (roots, stem, mid and upper branches)Dividing the figures 1 and 2 into sub-headings would allow to accurately describe the results, e.g. in line 152-153 (Fig.1A)153: In the Figure 1 there are, differences in the temporal dynamics in mid branches between xeric and mesic population .Figures 1 and 2 should contain information in which months the samples were taken:

stems monthly from April 2013 to March 2014, roots at two-month intervals from March 2013 to January 2014, the branches at two-month intervals from June 2013 to April 2014.

In the current form of the figure, the horizontal line represents the month (the month name should be clearly marked).

Line 164-166: For stem sugars or for branches sugars was observed higher sugar concentrations during colder winter months with shorter days than during warmer spring-summer months? (Figure 2). And  what was the difference between xeric and mesic population?

Author Response

Please see the attached point-by-point responses.
